# The Treatment of Pediatric Pain in Spain: A Survey Study

**DOI:** 10.3390/ijerph20032484

**Published:** 2023-01-30

**Authors:** Jordi Miró, Ester Solé, Elena Castarlenas, Pablo Ingelmo, Maria del Carme Nolla, Joaquín Escribano, Francisco Reinoso-Barbero

**Affiliations:** 1Universitat Rovira i Virgili, Carretera de Valls, 43007 Tarragona, Spain; 2Chair in Pediatric Pain, Unit for the Study and Treatment of Pain—ALGOS, Department of Psychology, Research Center for Behavior Assessment (CRAMC), 43007 Tarragona, Spain; 3Department of Anesthesia, McGill University, 1001 Boul. Decarie, Montreal, QC H4A 3J1, Canada; 4Xarxa Social i Sanitària, 43003 Tarragona, Spain; 5School of Medicine, Universitat Rovira i Virgili, IISPV, 43201 Reus, Spain; 6Department of Pediatrics, Hospital Universitari Sant Joan, Avgda. Dr. Josep Laporte 2, 43204 Reus, Spain; 7Pediatric Anesthesiology Service, Hospital Universitario La Paz, P. Castellana 261, 28046 Madrid, Spain

**Keywords:** adolescents, children, healthcare professionals, pain, pain management, pain programs

## Abstract

Pain is a common experience among children and adolescents, and pain management in this population is a challenge to clinicians. The aims of this study were to increase our understanding of current practices in the management of both acute and chronic pediatric pain in Spain, explore potential barriers to ideal practices, and identify professional needs as perceived by healthcare professionals. A total of 277 healthcare professionals took part, all of whom had wide experience in managing children and adolescents with pain (M [SD] age = 44.85, [10.73]; 75% women). Participants had to respond to a web-based survey with 50 questions related to pain education, organizational characteristics of their pain programs (including the characteristics of the patients treated), and current practices in the assessment and treatment of children and adolescents with pain. Almost all the participants (93%) acknowledged important gaps in their training, and only 47% reported that they had received specific education on the management of pediatric pain during their undergraduate and postgraduate studies. A third (31%) were members of multidisciplinary teams, and almost all (99%) understood that protocols to guide the management of pain in young people were necessary. However, only a few of them used a protocol to assess and treat (56% and 48%, respectively) acute and chronic pain (24% and 23%, respectively). The data also showed that a lack of pain education, coordination of professionals, and guidelines was perceived as an important barrier in the care provided to children and adolescents with pain in Spain. The findings of this study can now be used by healthcare professionals in Spain interested in managing pediatric pain, as well as policymakers concerned to improve the education of professionals and the care given to young people with pain.

## 1. Introduction

Pain is common among children and adolescents. For example, in the case of acute pain, research has shown that newborns undergo frequent stressful and painful procedures in neonatal intensive care units [1]. In addition, there is mounting evidence showing that the prevalence of moderate-to-severe pain while children are in hospital is high [2,3,4]. Similarly, research has also shown that the prevalence of chronic pain is also high [5] and increasing [6], including the most severe cases [7]. Pain can greatly interfere with the lives of young individuals and research indicates that children and adolescents with persistent pain report significant impairments in both physical function [8] and psychological function (i.e., emotional, cognitive and school/vocational; [9,10,11]).

The management of pain in children and adolescents has improved significantly in recent years [12]. However, breakthroughs in research are only occasionally reflected in daily clinical activity. For example, interdisciplinary pain treatment programs are considered the standard of care in the treatment of chronic pain [13]. However, they are only sometimes available and provided to patients [14]. Similarly, the treatment of acute pain is also complicated. For instance, although poly-pharmacological treatments are regularly prescribed to children and adolescents with pain [15], these treatments may not have been explicitly tested, there may be unanswered questions about their efficacy and safety [15,16,17], and they may even be contraindicated in pediatric populations [15,18]. 

It has been suggested that the limited access to adequate training and education of healthcare professionals is one of the main obstacles to proper pain management [19,20,21]. In a recent study of a representative sample of general practitioners and pediatricians working in Spain, all experts in the treatment of pediatric chronic pain, 83% of the participants stated that there were significant gaps in their training concerning pain management. Moreover, almost all of them (96%) identified this limitation in training as the most critical barrier to effective pain management [22]. However, this study only evaluated the participants’ perceptions of chronic pain, and it focused on only two profiles of medical professionals, leaving out other influential professionals usually involved in treating children and adolescents with pain, such as nurses, physiotherapists or psychologists. Increasing our understanding of how young people with acute and chronic pain are treated is key to improving both current treatment practices [14] and the training and education healthcare professionals receive about pain [23]. Given these issues, this research aimed to study current practices in managing pediatric pain in Spain, explore potential barriers to ideal practice, and identify professional needs, as perceived by healthcare providers assisting children with acute and chronic pain. 

## 2. Methods

### 2.1. Procedure

We used a survey developed by researchers of the Chair in Pediatric Pain of Universitat Rovira i Virgili, which has not been published but is available on demand from the corresponding author of this study. This survey has been used in a previous study by Miró and colleagues [22]. The survey included 50 questions divided into five sections that could be completed in 15 minutes. In the first three sections, the participants were requested to describe their professional background and practice, provide information on their training, and finally describe their current professional activity in pediatric pain management. In the last two sections, the participants were asked about the barriers they perceived to good patient care and what they needed to provide this care. Before the survey was launched, it was tested by five healthcare professionals with considerable experience in managing pain in children. The survey was conducted, and the data collected, before the first COVID-19 lockdown in Spain on 14 March 2020. 

We contacted colleagues—expert clinicians in managing pain in children and adolescents—and asked them to participate. In addition, we also asked them to share the information about this study with other professionals who they thought would be interested in participating and contributing. Moreover, we contacted regional pain centers, and specialized programs to identify additional experts. Finally, we also contacted Spanish pediatric and pain societies to help identify further additional experts. As a result, potential participants received an email about the study and a link to the survey. When participants entered the survey, they found a detailed explanation of the study and the informed consent page. Participants had to express their consent by clicking “YES” in response to a question about consent in order to complete the anonymous survey. The Human Subjects Review Committee of the Universitat Rovira i Virgili approved all study procedures (ref.: SO44/0303).

### 2.2. Data Analysis

We used absolute (n) and relative (%) frequencies to describe the participants’ responses. The analysis was performed using a SPSS statistics package, version 23.0 (SPSS, Chicago, IL, USA). 

## 3. Results

A total of 277healthcare professionals assisting children and adolescents with pain in Spain completed the survey. Most participants were pediatricians (40%), nurses (18%), and anesthesiologists (17%), although some had other professional profiles (see Table 1). The majority were women (75%) in their mid-forties (mean age = 44.85; SD = 10.73; age range = 24–68) working in a public hospital (64%). The sample of healthcare professionals was heterogeneous in terms of the problems and age of the patients treated. Nevertheless, the participants had considerable experience treating children and adolescents with pain (mean number of years = 14.94; SD = 10.03).

### 3.1. Pain Education

About half of the participants (47%) reported that they had received formal education on the management of pediatric pain during their undergraduate or postgraduate training. Continuing professional training programs and postgraduate training were the most common programs providing pain management education (see Table 2). Almost all participants (94%) reported limitations or significant gaps in their training on pediatric pain. Most said that their main sources of information were consulting with their colleagues, reading research papers, or using internet tools. Almost all participants (97%) said they would like to receive additional training, and reported a particular interest in non-pharmacological and pharmacological techniques. The online formats were the most appreciated by participants (see Table 2).

### 3.2. Characteristics of Programs and Professional Activities

Almost one third of participants (31%) were members of a multidisciplinary team as part of their job. General pediatrics and anesthesiology were the most frequent specialties in the teams (see Table 3). About half of the participants reported using protocols to assess (56%) and treat (48%) acute pain. However, less than a quarter of participants used protocols for assessing (24%) and treating (23%) chronic pain conditions. Coordination was one of the main problems identified by a significant group of professionals (33%) in this sample.

### 3.3. Current Practices in the Assessment and Treatment of Children and Adolescents with Pain

Most participants reported assessing pain intensity in children with both acute and chronic pain. The main reasons for not evaluating pain intensity were that they lacked the proper tools (16%) and the time to complete the evaluation (9%). The most evaluated domain in patients with chronic pain was pain intensity, followed by adverse effects and patient satisfaction. Most participants preferred easy-to-use scales to evaluate pain intensity (see Table 4).

Almost all participants (99%) reported that protocols to help pain management were necessary. However, only some reported that they followed one for either acute or chronic pain (48% and 23%, respectively).

Participants identified several pharmacological and non-pharmacological treatments implemented in their clinical practices. Although the specific treatments varied depending on the patient’s age and type of pain, Paracetamol and Ibuprofen were the most common pharmacological treatments. Heat and cold therapy and exercise were the primary non-pharmacological interventions (see Table 5 and Table 6).

### 3.4. Barriers to Practice and Professional Needs

Gaps in knowledge and the coordination of services were the most commonly perceived problem during the participants’ daily practice (see Table 3). The most critical problems in managing this population were the need for more information/education about managing pain in these patients, coordination with other professionals, and the little time available.

Developing multidisciplinary specialized units, improving communication, having clinical protocols, spending more time with patients and facilitating access to training courses were the actions suggested to improve clinical care.

## 4. Discussion

In this research, we studied the characteristics of current practices in Spain in the management of pediatric pain, explored potential barriers, and identified professional needs, as perceived by participants.

Four key findings emerged. First, only half of the participants received any formal training in pediatric pain management. Moreover, almost all reported significant gaps. This finding is similar to those in other reports that unanimously show that pain education needs to be improved at all levels, from undergraduate to specialized programs, and across disciplines [23,24,25,26,27,28], as education is essential, among other things, for effective practice. These results complement the findings from a previous study with pediatricians and family physicians working in Spain showing that the pain education of health professionals is limited [22]. Research has shown that significant hurdles to pediatric pain relief are the limited training of professionals and the small number of clinicians who are knowledgeable about pain management [29]. Although there is a great deal of information and a growing body of research on pediatric pain and its management, this information and knowledge are only sometimes used to guide clinical practice [20]. Finally, almost all participants would like additional training and most of them preferred internet-based platforms. It has been suggested that part of the problem with this less-than-ideal education is the education model. Pain-related content is often provided in individual lectures and included in modules that are not pain-specific [21]. A potential solution could be to create collaborative interprofessional pain education programs [30], including the patients’ view [31]. Research needs to be carried out to confirm the validity of these suggestions.

Second, organization, particularly the lack of coordination of healthcare professionals, was essential to this sample of participants. The data showed that less than one third of participants (31%) provide care as part of a multidisciplinary team, and a significant group (45%) reported that a less-than-optimal relationship with other professionals interfered with providing patients with the best treatment. This is particularly important in the case of chronic pain because the treatment of choice is multidisciplinary teams [12]. Standards for pain management services have been available for several years now [32]. However, the limited access to multidisciplinary pain treatments continues to be a significant challenge, even in developed countries such as Spain. Governments, researchers, clinicians, and patients’ advocacy organizations should all cooperate to ensure that these standards of care are available. In addition, organizations and individual professionals should cooperate to ensure that these standards of care are available to pediatric patients. 

Third, almost all participants advocated evidence-based guidelines to support their daily practice. However, in this study, only some participants reported using specific protocols to assess and treat their patients. Evidence-based guidelines are essential in setting the highest standards of care to help healthcare professionals, patients and their families [12]. However, limitations on adherence to available guidelines, their translation to daily practice and acceptance have been associated with sub-optimal care [33,34]. Guidelines are key to both the assessment and treatment of patients with pain. Assessment of pain in young people is often challenging, particularly in non-verbal children and those with intellectual disabilities. For example, although most pain intensity scales provide somewhat similar information [35,36,37], their scores are not completely concordant [38] and may reflect more than pain intensity [39,40,41]. Therefore, using an evidence-based guide on the best questionnaires for assessing pediatric pain intensity would be extremely helpful to clinicians. However, healthcare providers sometimes consider that validated tools are of interest only for academic purposes or research, and that they are time-consuming. Therefore, it is not a surprise that clinicians tend to avoid them if the belief is that the clinical data collected during their routine practice are enough to indicate whether their patients are achieving treatment goals. However, this is not a good practice as it is exposed to bias. For one, they tend to remember only their best outcomes and forget the bad ones (i.e., recall bias). Moreover, patients tend to exaggerate outcomes in order to please, or not offend, their doctor (i.e., reporting bias). Using validated outcome instruments reduces bias, especially if they are self-administered or administered by a third party. Healthcare providers should be able to use a brief set of assessment instruments with minimal inconvenience, while practicing pain care and management.

Likewise, perhaps a more obvious example of the need for a guide is to be found in the treatment of chronic pain. Current treatment guidelines emphasize the use of non-pharmacological treatments in managing chronic pain, but these strategies were the least used by this group of healthcare experts, and in their programs. In addition, participants reported prescribing poly-pharmacological treatments not supported by evidence-based guidelines. However, the participants should not be blamed as there are few data on this. For example, there is currently no evidence supporting pharmacological treatments such as antidepressants, anticonvulsants, non-steroidal anti-inflammatory drugs (NSAIDs), or opioids (excluding Tapentadol) for the treatment of pain in children or adolescents. On the other hand, there is evidence that medications such as gabapentinoids, antidepressants, codeine and Tramadol cause harm. Moreover, the incidence of adverse effects is largely unknown. Clinicians can only follow what has been used in previous studies as well as the recommendations for treating adults and children [42]. Not surprisingly, almost all participants (99%) advocated evidence-based guidelines to support their daily practice. There is an urgent need for longitudinal and head-to-head research to help clinicians and patients. Meanwhile, the most obvious action is to use strategies with strong evidence of benefit and little evidence of harm.

Finally, participants identified ways of improving clinical care: namely, the development of multidisciplinary specialized units, improved communication, clinical protocols, more time with patients and easier access to training. In addition, they reported a list of barriers to be overcome if they were to provide the best treatment possible. Participants reported that the most significant barrier was the lack of information on managing chronic pain. The most reported need was access to pediatric pain management guidelines and additional training. However, these barriers were at different levels: professional staff, patients, and the system itself. Examples of potential patient-related barriers include the reluctance to take analgesics because of a fear of side effects [43], or attitudes towards specific treatments (e.g., if patients think that their problems can only be solved by a pill it is not likely that they will accept psychological inputs). Examples of system barriers may include ill-defined pain management standards or limited access to specialists. Finally, one example of staff-related barriers is inadequate training, which may result in improper interdisciplinary work or a lack of attention to pain-related issues. Research has found that education and attitudes are among the most important factors that influence pain management in health care [44,45]. Therefore, studies are needed to identify what healthcare students, pain educators and patients think is important in order that pain care can be improved.

This study has some limitations that should be considered when we interpret the findings. First, the sample was one of convenience, which may or may not be representative. However, the results are similar to those reported in other studies, including one in Spain with similar objectives [22], which provides some support for its validity. Second, the number of participants in each of the healthcare specialties was different with the result that we could not compare their responses or elucidate whether their perceptions were similar, equal or different. Therefore, additional studies with larger sample sizes are warranted. Third, this research did not study treatments in relation to diseases. Therefore, we cannot comment on whether the treatments used are in line with clinical guidelines or not. Similarly, we did not study the differences in treatment, if any, among specialized and non-specialized institutions, or the motives for using one or another treatment, including pharmacological treatments. Thus, future studies should examine these issues.

## 5. Conclusions

Despite these limitations, this study provides new information that helps to improve our understanding of how to improve the treatment provided to children and adolescents with pain. The findings can now be used by healthcare professionals in Spain interested in managing pediatric pain, and policymakers concerned with improving the education of professionals and the care given to youths with pain, as well as their families.

## Figures and Tables

**Table 1 ijerph-20-02484-t001:** Profession and working place of participants.

**Professionals (n, %)**			
	General practitioner	11	4
	General pediatrician	110	40
	Anesthesiologist	48	17
	Pediatric rheumatologist	1	0
	Psychologist	4	1
	Nurse	50	18
	Physical therapist	23	8
	Surgeon	2	1
	Physical medicine and rehabilitation physician	17	6
	Pediatric oncologist	4	1
	Emergency medicine	2	1
	Radiotherapist	3	1
	Medical anthropologist	1	0
	Nursing assistant	1	0
**Working place (n, %)**			
	Public hospital	176	64
	Private hospital	15	5
	Primary care	50	18
	Private practice	9	3
	Others (e.g., Special education center, University, Child Development and Early Care Center or Patients’ association)	27	10

**Table 2 ijerph-20-02484-t002:** Pain education.

	n (%)
Do you have specific education in pediatric pain management?	
Yes	129 (47)
No	148 (53)
Where did you get the training from?	
Undergraduate training	15 (5)
Medical residency	44 (16)
Professional courses and postgraduate training	61 (22)
Continuing professional training programs	82 (30)
Others	11 (4)
Where do you look for the information you need?	
Consulting other specialists or colleagues	178 (64)
Scientific literature	162 (59)
Online information	121 (44)
Specific books on pediatric pain	61 (22)
Do you have any shortcomings in your training on how to manage chronic pediatric pain?	
Yes	261 (94)
No	16 (6)
In what areas would you expand the training?	
Non-pharmacological treatments	199 (72)
Pharmacotherapy	182 (66)
Pain assessment	178 (64)
Impact of pain on the quality of life	142 (51)
Biological mechanisms of pain	84 (30)
Surgical procedures	75 (27)
Epidemiology of pain	44 (16)
What are the most convenient formats for the training?	
Online training platforms	172 (62)
Online master classes	74 (27)
Face-to-face sessions during the week	14 (5)
Face-to-face sessions on the weekend	15 (5)

**Table 3 ijerph-20-02484-t003:** Organizational characteristics.

	n (%)
Are you member of a multidisciplinary team for the management of pediatric chronic pain?	
Yes	192 (69)
No	192 (69)
What medical specialists are members of the multidisciplinary team? **	
Pediatric gastroenterology	7 (8)
Pediatric neurology	10 (11)
Physiotherapy	13 (15)
General pediatrics	62 (73)
Anesthesiology	48 (57)
Traumatology	24 (28)
Pediatric rheumatology	13 (15)
Psychology	21 (25)
Rehabilitation medicine	5 (6)
Nursing	5 (55)
Occupational therapy	3 (4)
Psychiatry	3 (4)
Surgery	2 (3)
What daily problems do you have in your clinical practice when managing chronic pain? n (%)	
Little time for each patient	83 (30)
Lack of specific training on how to handle pain	135 (49)
Coordination with other professionals is not adequate	92 (33)
Lack of specialized personnel	73 (26)
Lack of support from the healthcare system and the administration	64 (23)
Lack of material and/or scientific resources	37 (13)

Note: ** Data relative to the 85 participants who reported being a member of a multidisciplinary team. The percentages are calculated on the basis of these 85 participants. Participants were allowed to select more than one specialization.

**Table 4 ijerph-20-02484-t004:** Current practices in the assessment and management of children and adolescents with pain.

	n (%)
Do you assess pain intensity in children regularly?	
Never	9 (3)
Rarely	25 (9)
Sometimes	46 (17)
Almost always	113 (41)
Always	84 (30)
What are the main reasons for not evaluating pediatric pain? n (%)	
Lack of time	26 (9)
Lack of adequate tools	43 (16)
What are the most important characteristics when choosing a scale to assess pain?	
Easy-to-use	245 (88)
Good psychometric properties	68 (25)
Preferred by patients	53 (19)
What domains do you assess in children with chronic pain? **	
Pain intensity	172 (85)
Symptoms and side effects of treatment	108 (54)
Physical functioning of the child	85 (42)
Emotional response of the child	83 (41)
Overall satisfaction with the treatment	108 (54)
Social functioning of the child	65 (32)
Sleep	101 (50)
Fatigue	52 (26)
I do not usually assess these domains	15 (7)
Others	3 (1)
Do you use a specific protocol for treating chronic pain?	
Yes	63 (23)
No	159 (57)

** Note: Data on the 202 participants who reported treating children with chronic pain. The percentages are calculated on the basis of these 202 participants.

**Table 5 ijerph-20-02484-t005:** Pharmacological treatments used for acute and chronic pain *.

	Acute Pain						Chronic Pain					
	0–3 Years		4–6 Years		0–17 Years		0–3 Years		4–6 Years		0–17 Years	
Prescription	%	n	%	n	%	n	%	n	%	n	%	n
**Non-Steroidal Anti-Inflammatory Drugs**												
Acetaminophen	90	161	88	159	93	167	74	71	79	76	91	87
Ibuprofen	72	129	83	149	86	155	59	57	69	66	83	80
Diclofenac	3	6	4	7	19	34	1	1	7	7	23	22
Naproxen	2	4	2	4	21	38	3	3	9	9	28	27
Acetyl-salicylic acid	0	0	0	0	1	2	0	0	0	0	2	2
Dypirone (metamizole)	23	39	28	47	33	55	9	9	13	13	17	16
**Opiate Drugs**												
Morphine	39	70	38	69	43	77	33	32	34	33	38	36
Tramadol	18	33	24	44	38	68	11	11	17	16	37	35
Fentanyl	37	67	38	69	42	75	20	19	23	22	28	27
Codeine	4	7	3	5	11	19	5	5	10	10	24	23
Methadone	6	11	6	10	7	13	9	9	12	11	13	12
Oxycodone	1	2	2	3	3	5	4	4	8	8	12	11
Hydromorphone	0	0	0.6	1	0.6	1	0	0	2	2	3	3
Tapentadol	0.6	1	0.6	1	1	2	2	2	1	1	5	5
**Co-analgesic Drugs**												
Steroids	18	33	18	33	26	47	24	23	31	30	33	32
Biphosphonates	0	0	0.6	1	3	5	0	0	5	5	8	8
Amitriptyline	0	0	3	5	7	13	5	5	13	12	21	20
Nortriptyline	0	0	0	0	0.6	1	0	0	0	0	3	3
Fluoxetine	0	0	0	0	0.6	1	1	1	2	2	7	7
Carbamazepine	0.6	1	1	2	4	7	1	1	6	6	10	10
Gabapentin	6	10	1	19	16	29	17	16	27	26	45	43
Pregabalin	0.6	1	2	4	8	15	5	5	8	8	22	21
Lidocaine	8	15	8	15	11	20	5	5	7	7	12	11
Mexiletine	0	0	0	0	0	0	1	1	0	0	0	0
Ketamine	18	33	18	33	22	39	5	5	7	7	12	11
Baclofen	1	2	4	7	6	10	7	7	8	8	9	9
Benzodiazepines	14	25	18	32	21	37	16	15	25	24	27	26

* Note: Data based on 180 participants (65% of the sample) who were prescribed medication for acute pain, and 96 participants (35% of the sample) who were prescribed medication for chronic pain.

**Table 6 ijerph-20-02484-t006:** Non-pharmacological treatments used for acute and chronic pain *.

	Acute Pain						Chronic Pain					
	0–3 Years		4–6 Years		0–17 Years		0–3 Years		4–6 Years		0–17 Years	
Intervention	%	n	%	n	%	n	%	n	%	n	%	n
Surgery	3	5	4	8	5	10	4	4	7	7	9	10
Nerve blocks	21	39	22	40	23	43	11	12	16	17	23	25
Acupuncture	1	2	2	4	5	10	2	2	4	4	12	13
Therapeutic massage	25	46	29	54	31	57	33	36	35	38	39	42
Chiropractic	0.5	1	0.5	1	1	2	1	1	2	2	3	3
Osteopathic	3	6	2	4	6	11	6	6	9	10	8	9
Use of cold/heat	35	65	48	88	54	99	42	45	50	54	59	64
Physical exercises (e.g., stretching)	23	43	32	59	44	81	38	41	48	52	57	61
Transcutaneous electrical stimulation	3	6	6	11	17	31	4	4	8	9	24	26
Ultrasounds	4	7	5	9	13	23	7	8	11	12	19	21
Hydrotherapy	3	6	6	11	9	17	33	36	13	14	20	22
Distraction techniques	51	94	62	114	57	104	1	1	1	1	1	1
Reflexotherapy	0.5	1	2	4	4	8	3	3	2	2	4	4
Music therapy	2	3	2	3	4	7	2	2	2	2	2	2
Kangaroo method	42	78	0	0	0	0	0	0	0	0	0	0
Breastfeeding	59	108	0	0	0	0	0	0	0	0	0	0
Sucrose administration	50	92	0	0	0	0	0	0	0	0	0	0
Relaxation	0	0	0	0	0	0	1	1	2	2	2	2
Biofeedback	0	0	0	0	0	0	3	3	5	5	9	10
Hypnosis	0	0	0	0	2	3	0	0	1	1	4	4
Cognitive-Behavioral Therapy	0	0	0	0	0	0	11	12	27	29	37	40
Acceptance and Commitment Therapy	0	0	0	0	0	0	8	9	12	13	29	31

* Note: Data based on 184 participants (66% of the sample) who used non-pharmacological treatments for acute pain, and 108 participants (39% of the sample) who used non-pharmacological treatments for chronic pain.

## Data Availability

Data is unavailable. However, future readers may obtain additional information upon request from corresponding author.

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
