# Peer review of "The Treatment of Pediatric Pain in Spain: A Survey Study"

_ijerph, 2023, doi:10.3390/ijerph20032484_

Round 1

Reviewer 1 Report

The authors reported a Country Survey about the treatment of pediatric pain to increase their knowledge about the management of pediatric pain (acute and chronic). A small sample of 277 healthcare professionals with wide experience in the management of children and adolescents with pain in Spain has been enrolled.

Their reported that pain education, coordination of professionals, and guidelines were perceived as important barriers in the care provided to children and adolescents 30 with pain in Spain.

I appreciate the effort of the authors to propose a helpful survey to improve the knowledge.

After careful review, I would suggest some items to check:

- use the same font for the whole manuscript;

- English needs a revision;

- how to improve the management of pediatric pain care should be proposed by the author (based on data from this survey). 

Author Response

The authors reported a Country Survey about the treatment of pediatric pain to increase their knowledge about the management of pediatric pain (acute and chronic). A small sample of 277 healthcare professionals with wide experience in the management of children and adolescents with pain in Spain has been enrolled.

Their reported that pain education, coordination of professionals, and guidelines were perceived as important barriers in the care provided to children and adolescents 30 with pain in Spain.

I appreciate the effort of the authors to propose a helpful survey to improve the knowledge.

Authors’ response: We thank the reviewer for the kind comment.

After careful review, I would suggest some items to check:

- use the same font for the whole manuscript;

Authors’ response: Done as suggested.

- English needs a revision;

Authors’ response: The manuscript has been revised by a linguist that is a native English speaker and professional translator.

- how to improve the management of pediatric pain care should be proposed by the author (based on data from this survey). 

Authors’ response: We thank the reviewer for this suggestion. The objective of this study was not to propose how to improve the management of pediatric pain. Although there are several lessons that can be learned from the findings. These “lessons” could be now used by healthcare professional associations, policymakers and educators to improve pain education and treatment as related to children and adolescents. The findings show in the direction of improving (1) pain education, (2) coordination of professionals, and (3) the use of available guidelines (which in part might be related to pain education). In the manuscript we provide specific suggestions in relation to these important issues. Moreover, as we also mention in the Discussion: “studies are needed to identify what healthcare students, pain educators and patients think is important so that pain care can be improved.”  

Reviewer 2 Report

This manuscript deals with the training, knowledge and practical implementation of pain therapy in children and adolescents in Spain. Barriers for an ideal therapy and necessary prerequisites will be recorded. For this purpose, a questionnaire is used with therapists of different professions and their statements are analyzed.

The questions cover the training of the therapists, organizational structures and therapy characteristics in both, acute pain treatment and chronic pain therapy.

In summary, the survey confirms a significant deficit in structured pain medicine education, systematic use of assessments, and lack of structured therapy protocols. Practitioners lack the prerequisites and implementation of interdisciplinary therapy.

General comments

The topic is relevant for the practical implementation of pain therapy in children and adolescents. However, the topic is not new and parts of it have already been investigated and published by the same working group.

The methodology of a web-based questionnaire is basically appropriate and common. However, the evaluation includes only a quantitative analysis of the answers of all participants. A qualitative analysis of structures of training, workplace situation, pain management of different diseases and age groups, specialized and non-specialized institutions, interdisciplinary teams is not investigated or described. Also, the specific selection of participants in the survey, e.g. access via clinic portals, specialist societies or others remains open. Despite extensive data analysis and presentation of results, the gain in knowledge therefore remains comparatively low in respect to previous studies of the working group. The discussion is therefore more a position paper and call for improvement of structures, less a critical discussion of the results.

The mixing of acute pain therapy with chronic pain management using the same assessment seems unfortunate. Relationships and differences between these two groups are not always clear, both in the results and within the discussion.

The references include an unusually high proportion of citations from the same research group (self-citations), 15 of 48 citations, even though the author team has made a considerable scientific contribution to similar topics in pain management. Some references are incomplete, and some tables could be optimized.

Specific comments

Abstract

Line 17                „..277 healthcare professionals….“
                              see comment in methods, line 86 on the selection of the participants

Introduction

Line 93                Reference No. 7 seems illogical, the authors and the title of the reference correspond exactly to this present manuscript. More detailed information about the publication in J. Pain is not given. Publication rejected there?

Line 52 -54         Reference (15) refers to chronic pain disorders in children and adolescents, but the statement in the manuscript refers to acute pain in children and adolescents.

Methods

Line 75                The main reference to the study methodology is a survey of the working group. The information on reference no. 25 is incomplete, the publication is not listed in pubmed or google scholar. Therefore, the initial survey is not available for the readers.

In order to be able to transfer or compare the data and recommendations collected here to the situation in other countries in the future, the survey should be made available as a supplement and the specific questions should be comprehensible.

Line 86                The specific selection of participating pain therapists, different professions and so-called experts remains unclear. Were specific clinics, universities, pain centers approached? The presentation of the participants (Table 1) in results (line 108 - 109) suggests that participation was rather random. See also comment below in results (line 104 - 107).

Line 95-97          Only a quantitative analysis of the participant data is provided. A qualitative methodological analysis is not described.

Results

Line 104-107      „The sample of healthcare … was heterogenous …. vast experience…“

Table 1 below shows that predominantly physicians participated, non-physician disciplines are underrepresented, in particular only three psychologists from 2 centers participated. Apart from a minimal proportion of rheumatologists and oncologists, no further information is given here on the specializations of the pediatricians or the patient group treated. This contrasts with the data in Table 3, lines 129-130, where information on multidisciplinary teams is provided. The latter refers to 89 participants. However, the numbers in Table 3 exceed 85. Therefore, the representations in Table 1 and 3 are not well understood and require explanation.

                              No information is given on patient groups, underlying diseases or treatment foci. On the other hand, the results section provides information on acute pain treatments and chronic pain therapy. These presentations are partly parallel, partly separate. This is confusing.

Tab.3,

line 129-130       see comment above, to line 104-107
lower part of the table, text and number rows are not at the same height

line 149-150       „Heat and cold therapy … were the primary non-pharmacological interventions“
that is remarkable and unusual. Are these applications by physiotherapists or other professions? Are there any special indications, patient groups or is it for other reasons?

Line 151-152      The presentation of Table 5 is confusing. What does each number mean? Number of survey participants using such therapies for acute or chronic pain? This figure is not helpful. For acute pain, pharmacotherapy depends on disease, injury, post-operative treatment, use, dose, duration, inpatient or outpatient treatment, combinations vary greatly. Therefore, the information provided here may not reflect actual prescribing practices.

The use of non-opioid and opioid analgesics for chronic pain disorders is unusual or noteworthy. No evidence exists for their use in childhood and adolescence for these disorders. What explains this high use of these subtances? E.g., lack of non-drug therapy programs or other established approaches?

Tab. 4 and 5       Results of a purely quantitative analysis of which therapies are used show the methodological weakness of the survey. Any information on the qualitative use of therapy, dependence on patient group, diseases, availability, indications or the like is missing. Thus, an evaluation of pain therapy is not possible, neither in terms of assessments, protocols or training.

Chap. 3.4

Line 153-158      „barriers of practice…“
The survey can only confirm the basic problem. Since the selection of participants appears random and patient groups or indications are not known, this survey cannot provide any new insights.

Diskussion

Line 162-180      The results confirm the preliminary studies of the same working group from 2019. The question remains unclear, what does the survey bring in terms of new findings compared to previous studies of the working group (Ref No 7, 23 24, 25)?

Degree and extent of training, content, disease- or age-related aspects in the treatment of acute pain and therapy concepts of chronic pain diseases are not differentiated. A differentiated view of pain therapy in emergency care, intensive care, postoperative treatment or in specific disease groups of neonatology, chronic inflammatory diseases, treatment of children with severe multiple disabilities is necessary for training, practical application, development of protocols, assessments and guidelines. Here, the collaboration of many disciplines is necessary for the development of these programs in education, implementation in clinical practice, and the development of guidelines.

Unfortunately, the authors of the survey do not address this in their analysis. However, this seems useful for the discussion and the development of future research questions.

Line 243-245      „The most reported need  …. Itself“
This statement is obvious, but is not supported by the results.

                              In summary, therefore, the discussion is more of a position paper and a call for improvement of structures, rather than a critical discussion of the results.

                              The different aspects of the discussion, the evaluation of the results and their consequences should be more clearly contrasted.

Line 264              „… provide new information“
this is not clear taking into account the studies published to date on this subject.

References

                              The reference list requires careful revision. Numerous references are not clearly identified, e.g. Ref. No. 7, 20, 25, 33, 45.

The high number of self-citations by the working group is striking, 15 of the 48 references.

Author Response

General comments

The topic is relevant for the practical implementation of pain therapy in children and adolescents. However, the topic is not new and parts of it have already been investigated and published by the same working group.

Authors’ response: As mentioned in the text, this study contributes to available research, including our previous research, and knowledge, by (1) studying acute pain management (our previous study was limited to the study of chronic pain management) and (2) including additional healthcare professionals to the analysis (our previous study was limited to 2 professional profiles -medical staff, and particularly to pediatricians and family physicians-. However, in this study there are 10 professional profiles).

The methodology of a web-based questionnaire is basically appropriate and common. However, the evaluation includes only a quantitative analysis of the answers of all participants. A qualitative analysis of structures of training, workplace situation, pain management of different diseases and age groups, specialized and non-specialized institutions, interdisciplinary teams is not investigated or described.

Authors’ response: The reviewer is correct. This study provides data in relation to the current situation of pain management (both acute and chronic) in children and adolescents as perceived by healthcare professionals. In this study we did not ask for the details and characteristics that the reviewer suggests. Thus, we cannot comment on those. However, we concur with the reviewer that a qualitative analysis of structures, pain management of different diseases, etc. is of interest. Therefore, we have added this as an important issue to be addressed in future studies (see lines 269-273).

Also, the specific selection of participants in the survey, e.g. access via clinic portals, specialist societies or others remains open.

Authors’ response: We thank the reviewer for this comment. We realize that the description of the procedure needed to be improved. Therefore, in this revised manuscript we have edited the text for completeness and improved comprehension (see lines 88-93). 

Despite extensive data analysis and presentation of results, the gain in knowledge therefore remains comparatively low in respect to previous studies of the working group. The discussion is therefore more a position paper and call for improvement of structures, less a critical discussion of the results.

Authors’ response: As described in the response to the first comment of this reviewer, there are important issues that this research addresses that had not been studied before. Furthermore, these data can be now used to improve what is being done. Nevertheless, and as mentioned in a response to a comment by the other reviewer, we have revised the Discussion section to highlight how the data collected can be now used by healthcare professionals in Spain interested in managing pediatric pain, and policymakers concerned with improving the education of professionals and the care given to youths with pain and their families.

The mixing of acute pain therapy with chronic pain management using the same assessment seems unfortunate. Relationships and differences between these two groups are not always clear, both in the results and within the discussion.

Authors’ response: We have revised the document and edited the content, were appropriate, for clarification.

The references include an unusually high proportion of citations from the same research group (self-citations), 15 of 48 citations, even though the author team has made a considerable scientific contribution to similar topics in pain management.

Authors’ response: We have revised the references for adequacy of citations. In this revised manuscript, we have deleted two of the references.

 Some references are incomplete

Authors’ response: We apologize for this. We have revised the references and completed those that were incomplete (see References).

, and some tables could be optimized.

Authors’ response: We have revised and edited the tables for optimization (see Tables 1-6).

 Specific comments

 Abstract

Line 17                „..277 healthcare professionals….“
                              see comment in methods, line 86 on the selection of the participants

Authors’ response: It is unclear what was meant here. Please see our response in Methods, below.

Introduction

Line 93                Reference No. 7 seems illogical, the authors and the title of the reference correspond exactly to this present manuscript. More detailed information about the publication in J. Pain is not given. Publication rejected there?

Authors’ response: Ref 7 was incorrect, and we apologize for this. As shown, it was a mixture of two unrelated references, one including the names of the authors of this study (this piece was incorrect) and the journal title of the reference that should have been provided. It is unclear why this happened. Nevertheless, no publication was rejected, and we have corrected the reference. This reference is related to an epidemiology study that reports the percentage of children/adolescents that suffer from chronic pain and, by comparison, shows the increase of the problem in these years. The correct reference was/is: Miró J, Roman-Juan J, Sánchez-Rodríguez E, Solé E, Castarlenas E, Jensen MP. Chronic Pain and High Impact Chronic Pain in Children and Adolescents: A Cross-Sectional Study. J Pain. 2022. doi: 10.1016/j.jpain.2022.12.007.

Line 52 -54         Reference (15) refers to chronic pain disorders in children and adolescents, but the statement in the manuscript refers to acute pain in children and adolescents.

Authors’ response: We apologize for this. We have changed the reference, as it should have been in the first place.

Methods

Line 75                The main reference to the study methodology is a survey of the working group. The information on reference no. 25 is incomplete, the publication is not listed in pubmed or google scholar. Therefore, the initial survey is not available for the readers.

Authors’ response: Ref 25, as shown in the original submission, was related to the protocol of the study including the survey and questions. That is a working document that has not been published. However, it is available upon request from the corresponding author. This reference was included to clarify the precedence and availability of the survey. However, in hindsight, we realize that this was not of help. Therefore, in order to clarify this, in this revised version we have deleted this reference. In addition, we have revised and edited this piece for clarification, keeping the statement showing that the survey can be obtained upon request to the corresponding author (see lines 776-78).

In order to be able to transfer or compare the data and recommendations collected here to the situation in other countries in the future, the survey should be made available as a supplement and the specific questions should be comprehensible.

 Authors’ response: We decided not to add the survey to the paper but as we described in the text, the original Spanish version of the survey could be obtained, on demand, from the corresponding author (this information can be now found in lines 77 and 78 in this revised version). If the Editor considers that the original survey (which is in Spanish) should be made available to improve the understanding of the study we would be happy to do so.

Line 86                The specific selection of participating pain therapists, different professions and so-called experts remains unclear. Were specific clinics, universities, pain centers approached? The presentation of the participants (Table 1) in results (line 108 - 109) suggests that participation was rather random. See also comment below in results (line 104 - 107).

 Authors’ response: The selection of participants is described in page 2 Participants in this study were experienced clinicians and researchers. On the one hand, we asked our colleagues to participate, and share the information of this study with other colleagues. In addition, we asked  professional associations to share this information with their associates. Potential participants were sent a link to the survey study granted they fulfilled with inclusion criteria. We have revised and edited this piece for clarification (see lines 88-93).  

Line 95-97          Only a quantitative analysis of the participant data is provided. A qualitative methodological analysis is not described.

 Authors’ response: This is correct. The objectives of this study were not to provide a qualitative analysis. However, as we mentioned in relation to a previous comment of the reviewer, we have suggested this as an area of interest in future studies (see the Discussion section).

Results

Line 104-107      „The sample of healthcare … was heterogenous …. vast experience…“

Table 1 below shows that predominantly physicians participated, non-physician disciplines are underrepresented, in particular only three psychologists from 2 centers participated.

Authors’ response: This is correct: mostly physicians participated. However, it was 4 psychologists (there was a typo here; which means 1% of the sample; but previous figure “2” was not alluding to 2 centers but to 2% of the sample). This should be no surprise as the management of pain in children, and in Spain in particular, is essentially conducted by physicians. This information is significant, granted that we approached treatment programs and clinicians treating patients with chronic pain (and interdisciplinary treatment is the recommended treatment for these patients). The sample was heterogeneous, as 10 professional profiles participated.

Apart from a minimal proportion of rheumatologists and oncologists, no further information is given here on the specializations of the pediatricians or the patient group treated. This contrasts with the data in Table 3, lines 129-130, where information on multidisciplinary teams is provided. The latter refers to 89 participants. However, the numbers in Table 3 exceed 85. Therefore, the representations in Table 1 and 3 are not well understood and require explanation.

Authors’ response: Table 1 provides the information that participants shared with us in response to the survey.  It is correct that the information in Table 1 and Table 3 is different. In Table 1 we show the information provided by all participants in this study. Table 1 provides details of the participants in the study whereas in Table 3 we summarize the information provided by participants in relation to the professionals in their programs (these professionals are different from the ones participating in our survey). Participants were allowed to select more than one response in relation to questions 2 and 3 (this is why total numbers exceed 85). We have added a note to the Table for (see Table 3).

No information is given on patient groups, underlying diseases or treatment foci.

Authors’ response: In this study, we asked participants to identify the type of treatment provided in relation to acute and chronic pain, depending on the age of participants. The treatments related to specific underlying diseases or body locations was not the objective of this study.  However, we concur with the reviewer that this would be of interest, and have been added to the Discussion section as interesting lines for future research (see lines 269-273).

On the other hand, the results section provides information on acute pain treatments and chronic pain therapy. These presentations are partly parallel, partly separate. This is confusing.

Authors’ response:  In the Results section we highlight in the text some of the findings -those that we thought were of interest, while others are provided in the Tables to avoid unnecessary repetition and duplication. We have revised and edited the text for clarification.

Tab.3,line 129-130       see comment above, to line 104-107
lower part of the table, text and number rows are not at the same height

Authors’ response: This has been corrected.

line 149-150       „Heat and cold therapy … were the primary non-pharmacological interventions“
that is remarkable and unusual. Are these applications by physiotherapists or other professions? Are there any special indications, patient groups or is it for other reasons?

Authors’ response: These are treatments used by physiotherapists, but we do not have information for what type of pain. As we mentioned in a previous response, we asked participants to report on the type of treatments used for acute and chronic pain depending on the ages of the patients.

Line 151-152      The presentation of Table 5 is confusing. What does each number mean? Number of survey participants using such therapies for acute or chronic pain? This figure is not helpful. For acute pain, pharmacotherapy depends on disease, injury, post-operative treatment, use, dose, duration, inpatient or outpatient treatment, combinations vary greatly. Therefore, the information provided here may not reflect actual prescribing practices.

Authors’ response: The figures on Table 5 and 6 are related to the number (N) and percentage (%) of participants using each procedure/technique/pharmacologic treatment that was listed in the survey. We agree that the data does not reflect prescribing practices for a specific problem, and have added this as a limitation to the study (see lines 269-273).

The use of non-opioid and opioid analgesics for chronic pain disorders is unusual or noteworthy. No evidence exists for their use in childhood and adolescence for these disorders. What explains this high use of these subtances? E.g., lack of non-drug therapy programs or other established approaches?

Authors’ response: This is correct. In fact, there is very limited evidence on most of the drugs listed in the Table, and this is commented in the Discussion (see lines 228-243).  The reason for participants using these or other drugs is not known to us, as the motives for their prescriptions were not asked to participants. Thus, we cannot respond to this otherwise very important question. We have added this as a limitation to the study, and suggested as an area of interest for future studies (see lines 269-273).

Tab. 4 and 5       Results of a purely quantitative analysis of which therapies are used show the methodological weakness of the survey. Any information on the qualitative use of therapy, dependence on patient group, diseases, availability, indications or the like is missing. Thus, an evaluation of pain therapy is not possible, neither in terms of assessments, protocols or training.

Authors’ response: The reviewer is correct, as we did not intend to evaluate pain therapy. The objective of the study was to describe pain management for pediatric pain in Spain, and the interferences/barriers as perceived by healthcare professionals.  A qualitative analysis of pain therapy for specific diseases would be of great interest but should be a matter for additional studies. The data collected provide future readers the bases to understand what the issues are. In addition, based on the findings, avenues about what to do to improve the situation. We did not intend to evaluate pain therapy, the level of analysis is not whether pain interventions are appropriate or not, or whether the assessment questionnaires are the most adequate ones. However, we do report, for example, that most participants select the questionnaires based on easiness of use rather than the psychometric properties of the questionnaires. We do not say that the selection should be based on data, and on what current guidelines suggest. Nevertheless, future readers could easily spot this as a problem suggesting that education should be improved.

Chap. 3.4 Line 153-158      „barriers of practice…“
The survey can only confirm the basic problem. Since the selection of participants appears random and patient groups or indications are not known, this survey cannot provide any new insights.

Authors’ response: The selection of participants was not random, as described in the manuscript. The data collected in this study adds to the existing findings on this particular issue, as described in our responses to previous comments.

Diskussion

Line 162-180      The results confirm the preliminary studies of the same working group from 2019.

Authors’ response: There must be somewhat of a misunderstanding here. Our 2019’s study was about the pain content in the curricula of healthcare professions undergraduate degrees, which is not the subject of this study. However, this study shows that most participants reported receiving little training in pain management, and this, in fact, is in line with our previous study, and with many other studies conducted in other countries showing the need to improve pain education of healthcare professionals. 

The question remains unclear, what does the survey bring in terms of new findings compared to previous studies of the working group (Ref No 7, 23 24, 25)?

Authors’ response: For the most part, the objectives, participants, and findings of this study are different from those of our previous studies.

Ref 7, as we mentioned in a response to a previous similar comment of this reviewer, was incorrect, and we apologize for this. As shown, it was a mixture of two unrelated references, one including the names of the authors of this study (this piece was incorrect) and the journal title of the reference that should have been there. As described before, Ref. 7 is related to an epidemiology study that reports the percentage of children/adolescents that suffer from chronic pain and, by comparison, shows the increase of the problem in these years. In this revised version, we have corrected the reference: Miró J, Roman-Juan J, Sánchez-Rodríguez E, Solé E, Castarlenas E, Jensen MP. Chronic Pain and High Impact Chronic Pain in Children and Adolescents: A Cross-Sectional Study. J Pain. 2022. doi: 10.1016/j.jpain.2022.12.007.

Ref 23 is related to a study that reports on the treatment of chronic pain as provided by pediatricians and primary care physicians. In this study, however, the objectives, participants, and findings go beyond that. As we have described in previous responses to similar comments from this reviewer, this study addresses the issue in relation to acute and chronic pain, and as perceived from a number of other professionals that are also involved in the treatment of this population (this study includes 10 different types of professionals).

Ref 24 reports on the content of curricula in public and private universities in Catalonia (Spain), which is not the objective of this study. Of course, it is related in a sense, but both studies are completely different, but complementary.

Ref 25 is the protocol of the study including the survey and questions that is a working document that has not been published. However, it is available upon request from the corresponding author. This reference was included to clarify the origin and availability of the survey. In this revised version, we have deleted this reference and edited the content for understanding (see lines 76-79).

Degree and extent of training, content, disease- or age-related aspects in the treatment of acute pain and therapy concepts of chronic pain diseases are not differentiated. A differentiated view of pain therapy in emergency care, intensive care, postoperative treatment or in specific disease groups of neonatology, chronic inflammatory diseases, treatment of children with severe multiple disabilities is necessary for training, practical application, development of protocols, assessments and guidelines. Here, the collaboration of many disciplines is necessary for the development of these programs in education, implementation in clinical practice, and the development of guidelines.

Unfortunately, the authors of the survey do not address this in their analysis. However, this seems useful for the discussion and the development of future research questions.

Authors’ response: The reviewer has a good point here. We concur that these are important issues. However, they were not among the objectives of this study and we do not have the data to comment these issues. Nevertheless, they could be addressed in a long series of different studies. Following the comment of the reviewer, we have added a suggestion for future studies in the Discussion (see lines 269-274).

Line 243-245      „The most reported need  …. Itself“
This statement is obvious, but is not supported by the results.

Authors’ response: The results support the statement, as participants reported the different problems/barriers they encountered when helping with children/adolescents with pain. See lines 161-167 and Table 3 for a description of the problems faced.

                              In summary, therefore, the discussion is more of a position paper and a call for improvement of structures, rather than a critical discussion of the results.

Authors’ response: We have revised the Discussion and edited where appropriate to make sure that it provides a clearer critical discussion of the results. As can be seen, we highlight and list the main findings of the study, and discuss them in relation to the findings reported by previous studies in the area.

                              The different aspects of the discussion, the evaluation of the results and their consequences should be more clearly contrasted.

Authors’ response: We have edited the text in the Discussion for clarification. Statements there are based on the data of this study.

Line 264              „… provide new information“
this is not clear taking into account the studies published to date on this subject.

Authors’ response: As described in previous responses to similar comments of this reviewer, this study provides new information that is complementary to findings from previous studies.

References

                              The reference list requires careful revision. Numerous references are not clearly identified, e.g. Ref. No. 7, 20, 25, 33, 45.

Authors’ response: We have revised the references for adequacy of citations, and completed those that were incomplete.